# Factors Associated with Anti-SARS-CoV-2 Vaccine Acceptance among Pregnant Women: Data from Outpatient Women Experiencing High-Risk Pregnancy

**DOI:** 10.3390/vaccines11020454

**Published:** 2023-02-16

**Authors:** Marianna Maranto, Giuseppe Gullo, Alessandra Bruno, Giuseppa Minutolo, Gaspare Cucinella, Antonio Maiorana, Alessandra Casuccio, Vincenzo Restivo

**Affiliations:** 1HCU Obstetrics and Gynecology, ARNAS Ospedale Civico Di Cristina-Benfratelli Hospital, 90127 Palermo, Italy; 2IVF Unit, Department of Obstetrics and Gynecology, Villa Sofia Cervello Hospital, University of Palermo, 90146 Palermo, Italy; 3Department of Health Promotion, Maternal and Infant Care, Internal Medicine and Medical Specialties (PROMISE) “G. D’Alessandro”, University of Palermo, 90127 Palermo, Italy

**Keywords:** high-risk pregnant women, anti-SARS-CoV-2 vaccination, COVID, vaccine acceptance, Health Belief Model, barrier, educational level, gynecologist, communication, vaccine hesitancy

## Abstract

Pregnant women are at higher risk of severe Coronavirus disease 2019 (COVID-19) complications than non-pregnant women. The initial exclusion of pregnant women from anti-SARS-CoV-2 vaccines clinical trials has caused a lack of conclusive data about safety and efficacy for this vulnerable population. This cross-sectional study aims to define the factors related to vaccination adherence in a sample of women experiencing high-risk pregnancies. The recruited women completed a questionnaire based on the Health Belief Model. Data were analyzed to evaluate the associations between socio-demographic variables and vaccination acceptance through descriptive, univariate and multivariate logistic analyses. Among the 233 women enrolled, 65.2% (n = 152) declared that they would accept the anti-SARS-CoV-2 vaccine. Multivariate analysis showed that vaccination acceptance was associated with a high educational level (aOR = 4.52, *p* = 0.001), a low perception of barriers to vaccination (aOR = 1.58, *p* = 0.005) and the gynecologist’s advice (aOR = 3.18, *p* = 0.01). About a third of pregnant women are still hesitant about the anti-SARS-CoV-2 vaccine, probably because of the conflicting information received from media, friends, acquaintances and health institutions. Determining factors linked to vaccine hesitancy among pregnant women is useful for creating vaccination strategies that increase vaccination uptake.

## 1. Introduction

Coronavirus disease 2019 (COVID-19) is an acute respiratory infectious disease that has become a major threat to global public health, especially since the spread of the novel Severe Acute Respiratory Syndrome Coronavirus 2 (SARS-CoV-2). SARS-CoV-2 is mainly spread via respiratory droplets and can cause a wide range of clinical cases: the infection can be asymptomatic, cause mild respiratory symptoms or require hospitalization and advanced medical care, as in case of severe pneumonia with respiratory insufficiency, which potentially leads to death [1]. The emergence of four SARS-CoV-2 variants of concern (VOCs) at the end of 2020 has raised apprehensions about a possible reduced performance of currently available vaccines. Following the World Health Organization (WHO)’s lead, these variants are referred to as Alpha, Beta, Gamma, Delta and Omicron. Omicron, the currently widespread dominant variant, was first detected in November 2021 [1]. The literature suggests that Omicron has an increased transmissibility compared to the Delta variant and that current vaccines are less effective against Omicron infection, but they still protect against hospitalization and severe disease. Severity of COVID-19 is associated with increased age and pre-existing or underlying medical conditions such as heart disease, diabetes, chronic respiratory disease and cancer [2,3].

Pregnant women with COVID-19 were identified as a vulnerable or higher risk population as well, since they are at increased risk of developing severe illness, hospitalization, admission to intensive care unit (ICU) and death compared with non-pregnant women [4,5,6]. A multinational observational study conducted in 18 countries showed that pregnant women with a diagnosis of COVID-19 had a greater risk of admission to the ICU or high-dependency unit (RR, 5.04; 95% CI, 3.13–8.10) and were 22 times more likely to die (RR, 22.3; 95% CI, 2.88–172) [7]. Furthermore, a review showed that pregnant women with COVID-19 are exposed to a higher risk of adverse fetal consequences, such as preterm delivery (OR 1.47, 95% CI 1.14–1.91), stillbirth (OR 2.84, 95% CI 1.25–6.45) and admission to the neonatal ICU (OR 4.89, 95% CI 1.87–12.81) compared with pregnant uninfected women [4], while asymptomatic women had rates that were similar to the uninfected [8].

Vaccination seems to be the most cost-effective strategy to prevent adverse outcomes for the mother and fetus in case of infection from SARS-CoV-2. Anti-SARS-CoV-2 vaccination coverage among pregnant women was initially lower than in the general population of the same age. The main reason could be that the first official recommendation to vaccinate pregnant women was too vague, due to the lack of a position statement and explicit recommendations from scientific societies [9]. Furthermore, the exclusion of pregnant and breastfeeding women from preauthorization clinical trials contributed to uncertainty about the immunogenicity and safety of anti-SARS-CoV-2 vaccines in these cohorts [10]. Because of this lack of evidence, at the beginning of January 2021, the WHO stated that vaccination should be made available to “pregnant women who are in those groups at high risk of exposure (healthcare workers (HCWs) or women with comorbidities), after a discussion of the risks and benefits of vaccination with their health care provider” [11]. A few days after this, the Italian Obstetric Surveillance System (ItOSS) and the Italian Institute of Health (ISS) published their first interim guidelines following the WHO recommendations [12]. Consequently, HCWs had to assess the balance between potential risk and benefits on an individual basis, and advocated for vaccination only for breastfeeding and pregnant women at high risk of exposure to the virus, such as women who work in the healthcare front line or those who have comorbidities. Afterwards, on 22 September 2021, the ISS updated its guidelines and established that pregnant and lactating women are eligible to receive anti-SARS-CoV-2 vaccines, since studies have shown that their safety profiles are similar to non-pregnant and non-lactating individuals. On 13 December 2021, the ISS further updated the indications and recommended that the booster dose be offered to pregnant women who are in the second and third trimester and desire to be vaccinated [13,14]. On 7 October 2022, the Italian Health Ministry announced the approval of the fourth anti-SARS-CoV-2 vaccine dose for pregnant women [13].

In the meanwhile, inadvertent exposure to mRNA vaccination during pregnancy, reported in some clinical trials where women were unaware that they were pregnant, has contributed to filling the knowledge gap on the safety and efficacy of anti-SARS-CoV-2 vaccines [15,16,17]. It was only in February 2021 that the first phase of a 2/3 randomized, placebo-controlled, observer-blind vaccine trial including healthy pregnant women (Pfizer–BioNTech, ClinicalTrials.gov identifier: NCT04754594) was begun. Later, evidence emerged supporting the safety and efficacy of anti-SARS-CoV-2 vaccination during pregnancy [17], leading to increasing support for vaccination of pregnant and lactating women. A large observational cohort study of pregnant women highlighted that the estimated vaccine effectiveness 7–56 days after the second dose was 96% for any documented infection, 97% for symptomatic infection and 89% for COVID-19-related hospitalization [18]. Preliminary findings from three U.S. vaccine safety monitoring systems have so far not shown any safety signals among pregnant women who received anti-SARS-CoV-2 mRNA vaccines [17].

Vaccine-induced immunity in pregnant and lactating women and their newborns has been widely assessed: immune transfer of neutralizing anti-Spike IgG and T-cells from the mother to the newborn occurs through the placenta and breastmilk, which foster the breastfeeding infant’s developing immune system, suggesting a potential prevention of infection [10,19,20,21]. Current studies show that the pregnancy trimester does not affect SARS-CoV-2 antibody production [22].

There is a paucity of literature about anti-SARS-CoV-2 vaccination attitudes among women who are currently experiencing a pregnancy at high risk of obstetrical complications. The purpose of this study is to examine the acceptability of anti-SARS-CoV-2 vaccination during pregnancy, and factors associated with it, in a cohort of high-risk pregnant women through a cognitive-behavioral model.

## 2. Materials and Methods

A cross-sectional study was conducted from October 2021 to December 2021, and the consecutive recruitment was planned in the high-risk pregnancy outpatient services of two hospitals in Palermo, Italy. During the study period, 331 women attended the two facilities for pregnant women at high risk of obstetric complications. Pregnant women were recruited based on the following inclusion criteria: women aged 18 years or above; high-risk pregnant women who attended antenatal care in the two participating hospitals; and participants who gave their written informed consent to participate in the study following the explanation of the study’s purpose. Afterwards, the participants completed an anonymous and self-administered paper-based questionnaire, which was developed using a previous questionnaire [23], and it was validated in a convenience sample of pregnant women.

The questionnaire was divided into three parts: the first one concerning socio-demographic factors, the second one concerning health status, and the third one about attitudes toward acceptance of anti-SARS-CoV-2 vaccination, including health beliefs related to SARS-CoV-2 infection and vaccination. Socio-demographic factors included: age, citizenship, place of residence, marital status, education and occupation. Health status included: parity (cumulative number of pregnancies, as reported by the participant), chronic diseases (diagnosed before pregnancy) and gestational complications (referring to diseases diagnosed during the current pregnancy such as gestational diabetes mellitus, gestational hypertension, gestational thyroid disorder and gestational anemia). The third part focused on anti-SARS-CoV-2 vaccination, investigating vaccination uptake, vaccine willingness, sources of information about anti-SARS-CoV-2 vaccination and a section related to the Health Belief Model (HBM) on vaccination acceptance. The acceptance of the anti-SARS-CoV-2 vaccine was evaluated according to the answer to the question “Are you willing to take the anti-SARS-CoV-2 vaccine?”. Furthermore, participants were queried about their sources of information on the anti-SARS-CoV-2 vaccine (web consultation, books, newspapers, gynecologist, general practitioner (GP), television, other practitioners, friends and family). To further assess the factors related to attitudes toward the anti-SARS-CoV-2 vaccine, the behavioral-cognitive attitude was analyzed. Several theoretical frameworks have been applied to understand and predict vaccine intention and behavior, such as the Protection Motivation Theory (PMT), the Theory of Planned Behavior (TPB) and the HBM [24]. The HBM theorizes that individuals are more willing to adopt behaviors in order to prevent diseases (such as vaccination) if they perceive that they are at increased risk to get a disease (perceived susceptibility), the disease and its consequences are harmful (perceived severity), the barriers to overcome in order to perform the behavior are minimal (perceived barriers) and the behavior is effective in reducing the risk of getting a disease (perceived benefits) [24].

The questionnaire used in this study adopted the HBM, using 8 questions: perceived susceptibility to COVID-19 for mother and infant (2 items), perceived severity of COVID-19 for mother and infant (2 items), perceived barriers to anti-SARS-CoV-2 vaccination (2 items) and benefits of anti-SARS-CoV-2 vaccination (2 items). The answers to the HBM were assessed using a 5-point Likert scale, ranging from “1—Strongly disagree” to “5—Strongly agree”. For each respondent, the scores relating to the 2 items of the same domain were added (range score 2–10 per domain) and the median value was calculated (median score = 8). The median value was used as a cutoff to recategorize the score for each domain as “High level” ≥ 8 or “Medium-Low level” < 8.

The study was conducted in accordance with the Declaration of Helsinki, and approved by the Ethics Committee Palermo 1 (protocol code 11/2020 at 18/12/2020).

### Statistical Analysis

The sample size (n = 213) was estimated using an expected prevalence of anti-SARS-CoV-2 vaccine coverage of 60%, a confidence level of 95%, and an expected refusal of 10%. Socio-demographic characteristics and vaccination acceptance of all the recruited pregnant women were summarized using frequencies and percentages. To evaluate the distribution of quantitative variables such as age, the skewness and kurtosis test was performed. Mean and standard deviation (SD) were chosen for the normal distribution of these variables, while median and interquartile range (IQR) were used for the non-normal distribution. The differences in quantitative variables normally and not normally distributed among vaccination acceptance were evaluated, respectively, with the Student *t* test and with the Wilcoxon and Mann Whitney test; while for the qualitative variables, the Chi2 test was used.

Bivariable analyses were performed to assess the associations between factors hypothesized to be associated with vaccination acceptance, including socio-demographic characteristics, health status and HBM items (Odds ratio (OR) with a confidence interval of 95%). The significant (*p* < 0.05) factors associated in bivariable analyses were run into a multivariable logistic regression model in order to identify predictors of vaccination and intention to be vaccinated. A *p*-value of <0.05 was considered statistically significant. Statistical analyses were performed using Stata/SE 14.2 (Copyright 1985–2015, StataCorp LLC, 4905 Lakeway Drive, College Station, TX 77845, U.S. Revision 29 January 2018).

## 3. Results

Overall, 237 women were eligible for the study, and 233 (98.3%) filled out the questionnaire. Table 1 describes the socio-demographic characteristics and health statuses of enrolled women. The mean age of the respondents was 32 years (SD ± 5.72). Pregnant women were more frequently Italian (97.5%; n = 227), currently living in the city of Palermo (59.6%; n = 139) and were at their first pregnancy (40.7%; n = 95). Concerning educational status, only a small proportion of respondents (21.4%, n = 50) had a college educational background or above, while two thirds of the participants (65.2%, n = 153) were unemployed (housewives and students). Regarding health status, enrolled women more frequently had more than one parity (59.3%; n = 138) and no chronic disease (70.8%; n = 165).

Regarding the anti-SARS-CoV-2 vaccine, two thirds of the sample (65.2%; n = 152) reported that they were willing to vaccinate. Women who accepted the anti-SARS-CoV-2 vaccine were more likely to have a bachelor’s degree than women who did not accept it (27% vs. 11.1%, *p* < 0.001).

Table 2 shows the answers related to the anti-SARS-CoV-2 vaccine and the HBM assumptions. Less than half of the enrolled women (46.8%, n = 109) had already received the anti-SARS-CoV-2 vaccine. The source of information on vaccination most frequently reported was the mass media (36%, n = 84), followed by a gynecologist (30.4%, n=71) and a GP (22.3%, n = 52). Among the people who accepted the anti-SARS-CoV-2 vaccine, their gynecologist was more frequently the source of information (38.1% vs. 16.1%, *p* < 0.001). Furthermore, regarding the susceptibility items of the HBM, pregnant women who accepted the anti-SARS-CoV-2 vaccine were more concerned about getting COVID-19 (strongly agree: 50% vs. 33.3%, *p* < 0.01). Similarly, those who accepted anti-SARS-CoV-2 vaccination were more concerned about the risk for their unborn babies to be infected by COVID-19 (strongly agree: 54% vs. 33.3%, *p* < 0.01). In the barriers dimension of the HBM, pregnant women who accepted vaccination were more likely to believe that the vaccination is safe during pregnancy and effective against SARS-CoV-2 (safety: strongly agree 28.3% vs. 7.4%, *p* < 0.01, effectiveness: strongly agree 36.8% vs. 3.7%, *p* < 0.01). In the benefits dimension, those who accepted the vaccination were more likely to strongly agree with the benefit of vaccination for themselves (41.4% vs. 6.2%, *p* < 0.01) and their fetuses (39.5% vs 6.2%, *p* < 0.01) than those who did not accept vaccination.

The bivariable analysis (Table 3) showed that having a bachelor’s degree or above (cOR 4.74; 95% CI 1.58–5.6), being at a first pregnancy (cOR 0.48; 95% CI 0.27–0.85), not reporting any pre-existing chronic disease (cOR 1.85; 95% CI 1.02–3.35) and having a gynecologist as a source of information (cOR 3.18, 95% CI 1.28–7.92) were statistically significant factors in association with vaccination acceptance.

The multivariable logistic regression model (Table 3), after adjusting for confounding variables, demonstrated that having a high school diploma (aOR 4.52; 95% CI 1.79–11.39), receiving a gynecologist’s advice (aOR 2.55; 95% CI 1.57–4.14), and a low level of perceived barriers to vaccination (aOR 0.63; 95% CI 0.45–0.86) were independently associated with anti-SARS-CoV-2 vaccine acceptance.

## 4. Discussion

This study aimed to assess women’s willingness to receive the anti-SARS-CoV-2 vaccine during a pregnancy at high risk of obstetric complications. Overall, anti-SARS-CoV-2 vaccine acceptance in this study (65.2%) was similar to that shown in a previous study (65%) conducted before the vaccines were approved for pregnant women [25]. Although evidence about safety and efficacy became clearer and the vaccination was openly endorsed by scientific societies, there is still not a full acceptance of anti-SARS-CoV-2 vaccine by pregnant women. The literature showed that anti-SARS-CoV-2 vaccine hesitancy was even more common during pregnancy, and that the main reason for vaccination refusal among pregnant women was the fear of potential consequences for their fetus or themselves [26]. The vaccine hesitancy was related not only to anti-SARS-CoV-2 vaccination [27], but also to seasonal influenza and pertussis vaccines, two long-lasting recommended maternal vaccinations [28].

One of the factors associated with anti-SARS-CoV-2 vaccine acceptance is a higher educational level. It has been already defined as a strong positive predictor of anti-SARS-CoV-2 vaccine acceptance in pregnant women, but also as a facilitator for pertussis and influenza vaccination during pregnancy [27]. A scoping review about COVID-19 vaccine hesitancy determinants in high-income countries has identified a lower educational level as a demographic characteristic that is strongly related to COVID-19 vaccine hesitancy [29]. This may be explained in different ways. First, a highly educated woman is likely to have better health knowledge, so she is more prone to perform healthy behaviors and make healthier lifestyle choices [30]. Second, a high educational level is often associated with a better economic status and, as a consequence, with greater access to quality healthcare and a high communication level with HCWs [31]. Third, a well-educated woman has greater access to different information sources, so she is less likely to be biased by inaccurate information about vaccines spreading on the Internet, leading to fewer probable potential misunderstandings about the safety of vaccines [32].

Another factor associated with anti-SARS-CoV-2 vaccine acceptance was the gynecologist as the main source of information. Literature has already shown that gynecologists’ recommendations have been critical means to encourage pregnant women to get vaccinated [33]. The gynecologist indeed has far more opportunities during prenatal care than the GP to advise pregnant women on whether to vaccinate. The gynecologist is the healthcare provider whom women trust the most, and is able to counter widespread misinformation about the anti-SARS-CoV-2 vaccine. In our study, about half of the participants received information from health care providers (GP, gynecologist, other HCWs), while the remaining ones personally acquired their knowledge from journals, television, and web consultations. Those findings are different from a previous study about maternal vaccinations, in which HCWs were reported as a source of information by three quarters of surveyed women [33]. This finding raises concerns about the role that media have been playing in spreading information about the anti-SARS-CoV-2 vaccine, as it has been proven that misinformation, both online and offline, lowers anti-SARS-CoV-2 vaccine confidence [34], and erroneous messages from one’s network of trust (e.g., friends and family) can create a barrier to vaccination [32]. Furthermore, the decline of HCWs as the main source of information also highlights that mistrust in the healthcare system and health institutions has been increasing in Italy since the beginning of the pandemic [35].

Furthermore, the HBM predicted vaccine acceptance in pregnancy. In relation to this model, the perception of barriers was another determinant associated with anti-SARS-CoV-2 vaccine approval in pregnancy. The finding that a high perception of barriers is negatively associated with a willingness to vaccinate is consistent with the results of other studies regarding vaccine-preventable diseases [36,37]. The present survey refers to “barriers” to the vaccination in pregnancy as those related to vaccine safety and the belief that the vaccine is not beneficial. Women who reject immunization, indeed, fear a potential threat to themselves or their fetus, since they are worried about the safety of vaccines, and believe that the benefits are darkened by the perceived negative maternal and fetal effects of vaccines [38]. To counteract this effect, communication strategies around the safety and effectiveness of the anti-SARS-CoV-2 vaccine in pregnancy should be implemented by HCWs that pregnant women trust, especially gynecologists.

In accordance with the benefit perception of the HBM, only half of the recruited women reported that they were aware that anti-SARS-CoV-2 vaccination in pregnant women protects the newborn in the first months of life in case of COVID-19 (moderately and strongly agree: 49.3%, n = 115). This low percentage may be due to the lack of appropriate counseling by the HCWs who meet women during their pregnancies about the benefits of conferring protection to the fetus and infant after birth. Public health decision-makers should promote effective interventions by healthcare providers, especially gynecologists and midwives, in order to implement vaccine uptake in pregnant women and target common misconceptions about maternal immunization [39]. A systematic review has highlighted a significant increase in vaccination uptake after interventions in pregnant women [40]. For example, a study offering onsite vaccination to pregnant women during antenatal visits in order to increase uptake showed a positive association between onsite availability of pertussis vaccine and maternal pertussis receipt during pregnancy [41]. Therefore, since the establishment of a trust-based relationship between a pregnant woman and her gynecologist is necessary for maternal immunization decision-making, these professionals have the responsibility of counseling pregnant women during antenatal visits in order to increase vaccination uptake and improve maternal and neonatal health.

Some limitations should be reported for the current study. First, it was based on the experiences of 233 women and, in spite of the heterogeneity of the sample, the analysis is representative of a single geographic area and not of the whole Italian obstetric population. Second, the survey was anonymous, so it was not possible to have access to medical and immunization records to validate self-reported vaccination status. Notwithstanding these limitations, the present survey is one of the first Italian studies on the anti-SARS-CoV-2 vaccine conducted before maternal visits. It provides important information about the factors associated with anti-SARS-CoV-2 vaccine acceptance.

## 5. Conclusions

Anti-SAR-CoV-2 vaccines were not fully accepted among the obstetric population. The reasons behind the low vaccination uptake were mainly to be found in the low level of knowledge about the disease and the lack of recommendations from healthcare providers, which caused doubts about vaccine safety, efficacy and benefits. It is necessary to understand the population’s vaccination barriers in order to improve maternal immunization on a local and national level more effectively, by providing tailored intervention programs for pregnant women.

## Figures and Tables

**Table 1 vaccines-11-00454-t001:** Socio-demographic characteristics and health statuses of pregnant women enrolled in the study about anti-SARS-CoV-2 vaccine acceptance.

	Total	Anti-SARS-CoV-2 Vaccine Acceptance		*p* Value
Socio-Demographic Characteristics	n = 233	Yes = 152	No = 81	
Age	32.1 [26.3–37.8]	32.6 [27–38.2]	31.2 [25.7–36.7]	0.09
Citizenship				
Italian	227 (97.5%)	148 (97.4%)	79 (97.5%)	0.94
Non-Italian	6 (2.5%)	4 (2.6%)	2 (2.5%)	
Area of residence ^1^				
Palermo	139 (59.6%)	94 (61.8%)	45 (55.5%)	0.18
Other cities than Palermo	92 (39.4%)	56 (36.8%)	36 (44.4%)	
Marital status				
Married	224 (96.2%)	146 (96%)	78 (96.2%)	0.92
Not married or divorced or separated	9 (3.8%)	6 (4%)	3 (3.8%)	
Educational level ^1^				
Less than high school diploma	100 (42.9%)	49 (32.7%)	51 (62.9%)	<0.001
High school diploma	81 (34.7%)	60 (40%)	21 (24.3%)	
Bachelor’s degree or above	50 (21.4%)	41 (27%)	9 (11.1%)	
Employment status ^1^				0.09
Unemployed or student or housewife	153 (65.6%)	95 (62.5%)	58 (73.4%)	
Employed	78 (33.4%)	57 (37.5%)	21 (26.6%)	
Health status				
Parity at enrolment				
One	95 (40.7%)	71 (46.7%)	24 (29.6%)	0.01
More than one	138 (59.3%)	81 (53.3%)	57 (70.4%)	
Gestational complications ^1^				0.74
Yes	35 (15%)	22 (14.7%)	13 (16%)	
No	193 (82.8%)	127 (85.3%)	66 (83.5%)	
Chronic diseases ^1^				0.04
Yes	64 (27.4%)	35 (23.5%)	29 (36.2%)	
No	165 (70.8%)	114 (76.5%)	51 (63.7%)	

^1^ Missing values: n. 2 (area of residence, educational level, employment status), n. 5 (gestational complications), n. 4 (chronic diseases).

**Table 2 vaccines-11-00454-t002:** Vaccination and Health Belief Model characteristics of the pregnant women by anti-SARS-CoV-2 vaccine acceptance.

	Total	Anti-SARS-CoV-2 Vaccine Acceptance		*p* Value
	n = 233	Yes = 152	No = 81	
SARS-CoV-2 vaccination ^1^				
Have you already received the anti-SARS-CoV-2 vaccine?				
Yes	109 (46.8%)	101 (66.5%)	8 (9.9%)	<0.001
No	122 (52.4%)	50 (32.9%)	72 (88.9%)	
Source of information on anti-SARS-CoV2 vaccine ^1,2^				
Nobody	18 (7.7%)	7 (4.6%)	11 (13.6%)	0.02
Media (television, web consultations)	84 (36%)	60 (39.4%)	20 (29.6%)	0.19
Books, newspapers	22 (9.4%)	15 (9.8%)	8 (9.8%)	0.74
Friends, acquaintances and family	20 (8.5%)	13 (8.5%)	7 (8.6%)	0.98
Gynecologist	71 (30.0%)	58 (38.1%)	13 (16.1%)	<0.001
GP	52 (22.3%)	32 (21.05%)	20 (24.7%)	0.52
Other physicians	13 (5.5%)	11 (7.2%)	2 (2.4%)	0.13
Other	6 (2.5%)	5 (3.3%)	1 (1.2%)	0.01
HBM items				
Perception of susceptibility				
Are you aware that you can get COVID-19?				
Strongly disagree	8 (3.4%)	1 (0.7%)	7 (8.6%)	<0.001
Disagree	15(6.4%)	4 (2.6%)	11 (13.6%)	
Neutral	50 (21.5%)	28 (18.4%)	22 (27.2%)	
Agree	57 (24.5%)	43 (28.3%)	14 (17.3%)	
Strongly agree	103 (44.2%)	76 (50.0%)	27 (33.3%)	
Are you aware that you can transmit the SARS-CoV2 to your fetus if you have COVID-19?				
Strongly disagree	18 (7.7%)	5 (3.3%)	13 (16.1%)	0.001
Disagree	18 (7.7%)	11 (7.2%)	7 (8.6%)	
Neutral	52 (22.3%)	28 (18.4%)	24 (29.7%)	
Agree	36 (15.5%)	26 (17.1%)	10 (12.3%)	
Strongly agree	109 (46.8%)	82 (54.0%)	27 (33.3%)	
Perception of severity				
If a pregnant woman gets COVID-19, is she more likely to have a severe disease?				
Strongly disagree	33 (14.1%)	16 (10.5%)	17 (21.0%)	0.08
Disagree	17 (7.3%)	9 (5.9%)	8 (9.9%)	
Neutral	77 (33.1%)	52 (34.2%)	25 (30.8%)	
Agree	58 (24.9%)	38 (25.0%)	20 (24.7%)	
Strongly agree	48 (20.6%)	37 (24.4%)	11 (13.6%)	
If a pregnant woman gets COVID-19, can the disease harm the unborn baby?				
Strongly disagree	25 (10.7%)	10 (6.6%)	15 (18.5%)	0.02
Disagree	29 (12.5%)	18 (11.9%)	11 (13.6%)	
Neutral	67 (28.7%)	42 (27.6%)	25 (30.9%)	
Agree	65 (27.9%)	45 (29.6%)	20 (24.7%)	
Strongly agree	47 (20.1%)	37 (24.3%)	10 (12.3%)	
Perception of barriers				
Is it safe to administer anti-SARS-CoV2 vaccine during pregnancy?				
Strongly disagree	41 (17.6%)	12 (7.9%)	29 (35.8%)	<0.001
Disagree	24 (10.3%)	11 (7.2%)	13 (16.1%)	
Neutral	79 (33.9%)	52 (34.2%)	27 (33.3%)	
Agree	40 (17.2%)	34 (22.4%)	6 (7.4%)	
Strongly agree	49 (21.0%)	43 (28.3%)	6 (7.4%)	
Is anti-SARS-CoV2 vaccine an effective measure to prevent COVID-19 in pregnant women?				
Strongly disagree	29 (12.4%)	7 (4.6%)	22 (27.2%)	<0.001
Disagree	23 (9.8%)	8 (5.3%)	15 (18.5%)	
Neutral	75 (32.9%)	44 (29.0%)	31 (38.3%)	
Agree	47 (20.1%)	37 (24.3%)	10 (12.3%)	
Strongly agree	59 (25.3%)	56 (36.8%)	3 (3.7%)	
Perception of benefits				
Is anti-SARS-CoV2 vaccine efficacious during pregnancy?				
Strongly disagree	29 (12.4%)	6 (3.9%)	23 (28.4%)	<0.001
Disagree	21 (9%)	8 (5.3%)	13 (16.0%)	
Neutral	69 (29.6%)	41 (27.0%)	28 (34.6%)	
Agree	46 (19.8%)	34 (22.4%)	12 (14.8%)	
Strongly agree	68 (29.2%)	63 (41.4%)	5 (6.2%)	
Can the SARS-CoV2 vaccination protect the newborn in the first months of life?				
Strongly disagree	30 (12.9%)	9 (5.9%)	21 (25.9%)	<0.001
Disagree	27 (11.6%)	7 (4.6%)	20 (24.7%)	
Neutral	61 (26.2%)	39 (25.7%)	22 (27.2%)	
Agree	50 (21.4%)	37 (24.3%)	13 (16.0%)	
Strongly agree	65 (27.9%)	60 (39.5%)	5 (6.2%)	

^1^ Missing values: n. 2 (SARS-CoV-2 vaccination), n. 21 (Source of information). ^2^ More than one answer was possible.

**Table 3 vaccines-11-00454-t003:** Bivariable and multivariable analyses of factors associated with anti-SARS-CoV-2 vaccine acceptance.

		Crude OR	CI 95%	*p*	Adjusted OR	CI95%	*p*
Age		1.04	0.99–1.09	0.09	0.95	0.88–1.02	0.19
Citizenship	Non-Italian	ref					
	Italian	0.93	0.16–5.22	0.94			
Area of residence	Palermo	ref					
	Other cities than Palermo	1.02	0.69–1.51	0.894			
Marital status	Not married or divorced or separated	ref					
	Married	0.93	0.22–3.84	0.93			
Educational level	Less than high school	ref			ref		
	High school diploma	2.96	1.57–5.60	0.001	4.52	1.79–11.39	0.001
	Bachelor’s degree or above	4.74	2.08–10.77	<0.001	4.35	0.98–19.26	0.052
Employment status	Unemployed or student or housewife	ref			ref		
	Employed	1.65	0.91–3.02	0.09	0.54	0.19–1.51	0.24
Health status							
Parity	One	ref			ref		
	More than one	0.48	0.27–0.85	0.01	0.49	0.22–1.10	0.08
Gestational complications	Yes	ref					
	No	1.13	0.53–2.40	0.34			
Chronic diseases	Yes	ref			ref		
	No	1.85	1.02–3.35	0.04	2.06	0.92–4.58	0.08
Source of information (yes vs no)	Nobody	0.30	0.11–0.82	0.02	0.41	0.10–1.69	0.22
	Media (television, web consultations)	1.49	0.81–2.74	0.19			
	Books, newspapers	0.85	0.33–2.15	0.73			
	Friends, acquaintances and family	0.98	0.37–2.58	0.98			
	Gynecologist	3.22	1.64–6.35	<0.001	3.18	1.28–7.92	0.01
	GP	0.81	0.43–1.54	0.52			
	Other physicians	3.08	0.66–14.25	0.15			
HBM items							
Perceived susceptibility	Low-Medium level	ref			ref		
	High level	1.37	1.2–1.57	<0.001	1.02	0.83–1.26	0.813
Perceived severity	Low-Medium level	ref			ref		
	High level	1.22	1.07–1.37	0.01	0.89	0.72–1.10	0.302
Perceived barriers	Low-Medium level	ref			ref		
	High level	0.59	0.5–0.68	<0.001	0.63	0.45–0.86	0.005
Perceived benefits	Low-Medium level	ref			ref		
	High level	1.63	1.42–1.88	<0.001	1.24	0.92–1.67	0.153

## Data Availability

Data will be available upon request sent to the corresponding author.

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
