# Peer review of "Factors Associated with Anti-SARS-CoV-2 Vaccine Acceptance among Pregnant Women: Data from Outpatient Women Experiencing High-Risk Pregnancy"

_vaccines, 2023, doi:10.3390/vaccines11020454_

Round 1

Reviewer 1 Report

Children and pregnant women are often the forgotten populations of clinical trials and were at first excluded from research on SARS-Cov2 studies. This resulted in a loss of chance for women to benefit from SARS-Cov2 vaccine in the early times of the pandemic. The later inclusion of the pregnant women who had high risk of severe outcome in vaccine trials and recommendation improved their conditions. However, there is still a lower rate of vaccine uptake in pregnant women lasting consequences of the initially diffusion of information concerns against vaccine safety. The purpose of this study was to examine acceptability of anti-SARS-CoV-2 vaccination during pregnancy and associated factors a cohort of pregnant women using cognitive-behavioral model. The paper showed that the vaccine acceptance depend of the higher educational level of the women and advices of the doctors.

Major comments

1) The authors listed “women at risk of pregnancy” has the main study population criteria. Authors need to define “women at risk of pregnancy” is it for any women attending the antenal clinic or those with confirmer pregnancy?

2) The study sample size calculation is not clearly specified. The authors should describe how their estimated the number of women included

3) In the results tables, the size of the study population in not always 233 women for all the variables. There are some missing data that need to be specified in table’s footnotes of the tables. Also, in table 2, the authors need to specify that items are not mutually exclusive for the question “Source of information on anti-SARS CoV2 vaccine”.

Minor comments

-          They are several writing errors for example “SARS-Cov-2” and not “SASR-Cov2” which need to be corrected in all the manuscript

-          The manuscript will need some editing ideally by an English expert.

Author Response

Major comments

1) The authors listed “women at risk of pregnancy” has the main study population criteria. Authors need to define “women at risk of pregnancy” is it for any women attending the antenal clinic or those with confirmer pregnancy?

Thank you for giving us the possibility to correct and clarify the population included in the study. The main study population is represented by high-risk obstetrical patients, so the pregnancy is confirmed and at high risk of obstetric complications.

2) The study sample size calculation is not clearly specified. The authors should describe how their estimated the number of women included.

We recruited a consecutive sample of patients attending the high-risk pregnancy antenatal clinic of two hospitals in the city of Palermo during the study period, that is from October 2021 to December 2021. Overall, 331 women attended the clinics during the study period.

3) In the results tables, the size of the study population in not always 233 women for all the variables. There are some missing data that need to be specified in table’s footnotes of the tables. Also, in table 2, the authors need to specify that items are not mutually exclusive for the question “Source of information on anti-SARS CoV2 vaccine”.

Thank you for the insightful suggestion. As you suggested, we have improved the tables with footnotes.

Minor comments

-They are several writing errors for example “SARS-Cov2” and not “SASR-Cov2” which need to be corrected in all the manuscript

Thank you for noting those mistakes. We have corrected it throughout all the manuscript.

-          The manuscript will need some editing ideally by an English expert.

Thank you for your suggestion. An English expert has edited the manuscript.

Reviewer 2 Report

Dear editor,

Thank you for the kind invitation to review this manuscript. It is generally well written and easy to follow

Title

- What does at risk pregnancy refer to?

Abstract

- Suggest to correct grammatical errors and sentence structuring issues

-> e.g the last sentence

Introduction

- The initial section on introduction appears to target the Omicron strain of COVID-19

- The section about vaccination in pregnancy can be more succinct and more literature should be highlighted about current evidence available.

-> There appears to be significant amount of literature related to vaccine hesitancy in pregnant ladies in pubmed contrary to the authors' description about paucity of literature on the matter. 

- A more detailed explanation needs to be made about the knowledge gap the authors hope to fill.

Methods

- What is the time period for the study (beyond the year) -> This is important given policies changes highlighted in this study during 2021.

- Were there any attempts to trial the questionnaire in any patient populations prior to its rollout

- How was the sample size determined for this study?

- It would be helpful to describe vaccination programs adopted in Italy during the study period especially pre and post policies changes. 

Results

- All acronyms should be defined before use

Minor comments

- A brief description of high vaccine hesitancy and healthcare workers should be described in the introduction.

-> https://pubmed.ncbi.nlm.nih.gov/34452026/

-> https://pubmed.ncbi.nlm.nih.gov/35455286/

Author Response

Title

- What does at risk pregnancy refer to?

 Thank you for giving us the possibility to correct and clarify the population studied. The main study population is represented by obstetrical patients with high-risk of obstetric complications, so the pregnancy is confirmed and at high risk of obstetric complications.”

Abstract

- Suggest to correct grammatical errors and sentence structuring issues

-> e.g the last sentence

We agree with you and have rewritten the whole abstract.

Introduction

- The initial section on introduction appears to target the Omicron strain of COVID-19

Thank you for your in-depth analysis, you have raised an important question. We focused on the Omicron strain because it was first detected when we were recruiting our sample and when the dose booster for pregnant women was first authorized. However, we edited the introduction to give more clarity about the VOCs.

- The section about vaccination in pregnancy can be more succinct and more literature should be highlighted about current evidence available.

We sincerely appreciate your insightful comment and we have modified the introduction using the last updates.

-> There appears to be significant amount of literature related to vaccine hesitancy in pregnant ladies in pubmed contrary to the authors' description about paucity of literature on the matter. 

This is an interesting perspective. Nevertheless, we were referring to the fact that we have surveyed high-risk obstetrical patients. Therefore, we have corrected it in the manuscript.

- A more detailed explanation needs to be made about the knowledge gap the authors hope to fill.

Thank you for providing these insights. As you suggested, we have provided more details about the important role of gynaecologists, who are in a unique position to educate and counsel patients throughout their pregnancies. In the conclusions we have underlined the necessity of tailored educational campaigns for pregnant women, which should be carried out by gynaecologists and midwives.

Methods

- What is the time period for the study (beyond the year) -> This is important given policies changes highlighted in this study during 2021.

I am sorry that this part was not clarified in the original manuscript. I should have explained that the study was conducted from October 2021 to December 2021, when the vaccination was just been approved for the whole obstetric population.

- Were there any attempts to trial the questionnaire in any patient populations prior to its rollout

That is an interesting query. The questionnaire was validated in a small sample of 10 pregnant women.

- How was the sample size determined for this study?

You have raised an important question. We recruited a consecutive sample of patients attending the high-risk pregnancy antenatal clinic of two hospitals in the city of Palermo during the study period, that is from October 2021 to December 2021. Overall, 331 women attended the clinics during the study period and 237 agreed to participate, but only 233 fulfilled the HBM section of the questionnaire.

- It would be helpful to describe vaccination programs adopted in Italy during the study period especially pre and post policies changes. 

I have improved the description of the timeline of the official vaccination recommendation from the Italian Institute of Health in pregnant women.

 Results

- All acronyms should be defined before use

Thank you for noting our mistake. We have corrected all acronyms.

 Minor comments

- A brief description of high vaccine hesitancy and healthcare workers should be described in the introduction.

-> https://pubmed.ncbi.nlm.nih.gov/34452026/

-> https://pubmed.ncbi.nlm.nih.gov/35455286/

Thank you for your interesting advice. We agree with you and have added this suggestion throughout our paper.

Round 2

Reviewer 2 Report

nil futher comments

Author Response

Thank you for your suggestions.